# When Self-Driving Fails: Evaluating Social Media Posts Regarding Problems and Misconceptions about Tesla's FSD Mode

**Anne Linja** [†] ⬤, **Tauseef Ibne Mamun** [†] ⬤ **and Shane T. Mueller** [*,†] ⬤

Department of Cognitive and Learning Sciences, Michigan Technological University, Houghton, MI 49931, USA
* Correspondence: shanem@mtu.edu
† These authors contributed equally to this work.

**Abstract:** With the recent deployment of the latest generation of Tesla's Full Self-Driving (FSD) mode, consumers are using semi-autonomous vehicles in both highway and residential driving for the first time. As a result, drivers are facing complex and unanticipated situations with an unproven technology, which is a central challenge for cooperative cognition. One way to support cooperative cognition in such situations is to inform and educate the user about potential limitations. Because these limitations are not always easily discovered, users have turned to the internet and social media to document their experiences, seek answers to questions they have, provide advice on features to others, and assist other drivers with less FSD experience. In this paper, we explore a novel approach to supporting cooperative cognition: Using social media posts can help characterize the limitations of the automation in order to get information about the limitations of the system and explanations and workarounds for how to deal with these limitations. Ultimately, our goal is to determine the kinds of problems being reported via social media that might be useful in helping users anticipate and develop a better mental model of an AI system that they rely on. To do so, we examine a corpus of social media posts about FSD problems to identify (1) the typical problems reported, (2) the kinds of explanations or answers provided by users, and (3) the feasibility of using such user-generated information to provide training and assistance for new drivers. The results reveal a number of limitations of the FSD system (e.g., lane-keeping and phantom braking) that may be anticipated by drivers, enabling them to predict and avoid the problems, thus allowing better mental models of the system and supporting cooperative cognition of the human-AI system in more situations.

**Keywords:** Explainable AI; Tesla FSD; user-centered AI; cooperative cognition

## 1. Introduction

Numerous companies are developing and attempting to field semi-autonomous and self-driving vehicles in the consumer market. Although the ultimate goal of many of these systems is to remove the human from the control and decision-making process, for the foreseeable future, a driver will remain in a supervisory role, taking over control when they deem that the vehicle is operating in an unsafe manner or is facing situations that it is not able to handle. Thus, both semi-autonomous and autonomous vehicles are multi-modal technologies whose interactions with humans involve a collaborative effort between the human and the artificial intelligence (AI) system, requiring monitoring and situational awareness on the part of the driver [1]. Cooperative cognition implies a give-and-take between the AI and the human, and although current self-driving vehicles have some limited inputs about the driver's state (e.g., whether they are attentive or have their hands on the steering wheel), cooperative cognition in these systems mostly remains in the form of ways to enable a user to develop a better and more accurate mental model of the AI system. That is, a user must develop an understanding of the AI system's training, capabilities, and expected failures in order to properly supervise the system. With a better mental model,

they can better anticipate problems, understand the relative consequences of the problems, and avoid the problems via takeover, routing decisions, or other interventions. We expect that the default assumption will be *cognitive anthropomorphism* [2]—assuming that the AI's cognitive abilities and decision-making patterns match those of a human driver. However, to the extent that they do not, this will impair a user's ability to anticipate problems, predict when it might be necessary to take over control from the AI, and avoid situations that might become dangerous. We suggest that for supervisory control problems in which the user has little or no ability to modify the AI, the main avenue for cooperative cognition involves the user developing a good mental model of the system and, thus, the ability to predict and recognize situations where they must take over.

One significant impediment to a user understanding the failures and limitations of the AI is that the developers of the semi-autonomous vehicles are often reluctant to advertise the boundaries and limitations of their systems. If they acknowledge these, they may both be subject to legal and regulatory consequences and generate reluctance on the part of potential customers, who may hesitate to buy a system that has safety or usability problems. Furthermore, although there are channels for reporting problems to both vehicle manufacturers and government agencies, these rarely result in direct feedback or advice to the person reporting the problem. Consequently, users have turned to third-party social media platforms to report problems to others in the community, to seek advice from others experiencing similar issues, to understand whether the problems are widespread, to see if there are workarounds for the issues, and to simply warn or caution others about specific situations.

When used by a community to help understand an AI system, social media platforms incorporate many of the elements of what Mamun et al. (2021) [3] described as Collaborative Explainable AI (CXAI)—a structured online tool to enable users to explain AI behavior to one another. This tool is framed as a non-algorithmic version of Explainable AI (XAI), insofar as it satisfies many of the goals that explainable algorithm developers have attempted to develop. In order to investigate the kinds of explanations and interactions involved in user-centered social media platforms regarding autonomous driving systems, we examined Tesla-related social media, collected posts, and conducted a detailed analysis of the problems reported on social media by users. In this paper, we describe the most common failures and limitations of the system and examine these communications in terms of their role in supporting collaborative explanation of the AI system. Finally, we relate these analyses to how they support the expectations and mental models of the users of the system and their prospects for supporting improved cooperative cognition through training.

### 1.1. Tesla's Self-Driving Modes

Among at least 14 companies developing semi-autonomous and autonomous driving systems, Tesla is perhaps the best known, having fielded at least two systems that support Level 2 partial vehicle automation [4] in their vehicles: Tesla Autopilot (AP) and Full Self-Driving (FSD) mode. Tesla AP was available to users starting in 2015 [5], focusing on highway driving situations. It featured traffic-aware cruise control and auto-steering, but the driver was still ultimately responsible for the control of the car. Perhaps ironically, Autopilot has been promoted as a safety solution [6], touted as abating potentially dangerous driving situations for humans and decreasing the number of accidents per mile on highways (https://www.tesla.com/VehicleSafetyReport, accessed on 1 February 2022). According to Tesla's website, Autopilot enables the Tesla vehicle to steer, accelerate, and brake automatically within its lane, all with human driver supervision.

Currently, Autopilot is a part of the more capable FSD system [7]. FSD offers more advanced driver assistance features than AP and allows assisted driving under the driver's active guidance, with the goal of enabling control in many more situations, such as residential and city driving. In October 2020, Tesla initially released a beta version of its FSD

software to a small group of users in the USA [8], and in October 2021, the system entered wider availability to more users.

In contrast to fully autonomous test cars (which are still mostly experimental), Tesla's semi-autonomous AP is already widely used by drivers. Reports suggest that fully autonomous test vehicles have driven several million miles on American roads, but Tesla AP users had driven 140 million miles worldwide by 2016 [9], and there were estimated to be nearly 100,000 FSD Beta users by April 2022 [10], so the numbers of potential users, performance problems, accidents, and incidents are much larger.

News articles have reported a number of problems faced by drivers using FSD, including difficulty with the Tesla identifying emergency vehicles with flashing lights, flares, illuminated arrow boards, or traffic cones near them [11]. In May 2022, the National Transportation Safety Board (NTSB) revealed that more than 750 Tesla owners had reported issues with FSD, including phantom braking and crashes; furthermore, news reports have discussed the possibility of recalls in order to deal with particular problems reported to the government safety boards. Thus, some of the problems with the FSD are being publicized, but these reports are channelled to government safety board reports that are not easily or immediately accessible to users and do not help those originally reporting the problem, as they may mostly assist the government in recommending recalls or Tesla in improving future versions of the software.

*1.2. Human Factors Research on Tesla Self-Driving Modes and Other Semi-Autonomous Vehicle Systems*

One way to help identify errors, failures, and problems in an automated system is through careful human factors and safety testing. Some of the limitations of the FSD mode have indeed been discovered through careful research evaluating the system with human users. There have been numerous studies evaluating AP, examining various aspects of human–technology integration and identifying the current gaps in knowledge in this domain. For example, Endsley [1] analyzed the first six months of her own driving of a Tesla Model S. She noted that although Tesla's service representative initially provided useful knowledge, the breadth of experiences that a driver needs in order to become an expert is wide ranging, and learning about the successes and failures of the technology can be ad hoc and must come from multiple sources.

The Tesla-specific research is also informed by results on how to improve the human–machine interaction in other (real and simulated) semi-autonomous vehicles. Overall, this research suggests that many warnings, alerts, and displays can be used to improve the safety and experience of the drivers, especially during takeover requests. For example, Figalova et al. [12] found that takeover requests made by the vehicle communicated via in-vehicle ambient light cues led to drivers being more prepared for the takeover, and they did so without adding to the driver's mental workload. Another study reported that an explanatory windshield-based augmented reality display [13] led to increased situational awareness. Thus, interface design can certainly improve user experience in these systems.

Other research has focused on how drivers' knowledge and expectations impact their use of these systems. One study [14] found that it took drivers approximately 30 h (or two weeks) to become accustomed to and proficient in autonomous driving, suggesting that it should be expected that users need to learn about and develop a reasonable mental model of the autonomous driving system. Research has shown that the typical driver of an autonomous vehicle has a large amount of driving experience, is confident with their computer expertise, and is interested in how automation works [15]—suggesting that current users are likely to embrace and try to understand the technology that they are using. A consequence of greater understanding of the technology is that it should garner a higher level of trust from users [16]. This is reinforced by Ruijten et al.'s [17] suggestion that vehicles should mimic human behavior, which may help to increase people's trust and acceptance of autonomous vehicles—vehicles that are naturally understood may be trusted more. This research suggests that an important aspect of the human factors of

operating autonomous vehicles is the human's mental model and understanding of the system. This understanding is mostly gained via first-hand experience, but it suggests that other ways of gaining that experience, including the shared experiences of other users, may also provide benefits.

### 1.3. Social Q&A Sites as a Repository of Tesla AI Failure Modes

So far, we have documented how some errors in self-driving capabilities get reported to government safety boards and the manufacturer, which are not easy for users to access. However, human factors research has suggested that a better mental model and clearer understanding of these limitations will help users anticipate and avoid potentially dangerous situations, which is the cornerstone of cooperative cognition in supervisory control. Detailed human factors evaluations have their limits because they are time-consuming and idiosyncratic. Lessons learned by individual drivers help them make themselves better, but do little for others. Thus, many of the ways in which users could develop better knowledge of the vehicle control system are difficult to access or generate. We suggest that social media can indeed play a supporting role in solving this problem, as a low-cost user-generated body of information that can both help users understand their own problems and educate other readers about situations that they have not yet experienced in order to anticipate and avoid future problems.

Social media platforms that support such interactions are generally referred to as Social Q&A (SQA)—a blanket term that refers to social media platforms in which people ask, answer, and rate question/answer content [18]. SQA platforms are typically public, community-based, reliant on freeform natural-language text [19] (rather than structured forms), and use simple voting schemes (rather than complex algorithms) to identify salient, relevant, and accurate information. When learning any new system, users can use SQA in two ways—actively, by posting questions or observations in order to get an explanation, or passively, by reading or searching for other users' accounts of problems. In both cases, even though the problems and solutions are potentially incorrect and are often posted by anonymous users, they can nevertheless be trustworthy because they are vetted by a community and are not being whitewashed by the system's developer.

To succeed, however, users of an SQA platform need to be sufficiently motivated to interact with the SQA platform. A small community or team may be motivated to communicate intrinsically, but other SQA systems have incorporated specific features that encourage contributions. Responders' authority, shorter response time, and greater answer length are some critical features that are positively associated with the peer-judged answer quality in an SQA site [20]. In the case of SQA related to autonomous AI, we also suggest that users are likely to be motivated by emotional responses to situations that they experienced—from minor feelings of annoyance or betrayal to anger or fear about deadly situations.

Some research has found that social forums can indeed be effective in realigning trust or distrust in an AI system. Koskinen et al. [16] reported that users of Tesla AP realigned trust in the system after misplacing trust when encountering unexpected situations that differed from their initial expectations. Social groups are also a tool to calibrate expectations and teach appropriate use of automation that can lead to fewer safety incidents through communication guidelines [21]. Social forums can also be used by researchers to learn more about safety issues that can only be discovered by users with specialized knowledge, unique orientations toward a subject, or experience in special circumstances [22].

Although the idea of giving explanations through social forums or social Q&A sites is still a relatively novel one, we previously argued [23] that the contents of a social Q&A site satisfy many of the 'Goodness Criteria' [24] of AI explanations. Thus, the use of social media will likely be helpful in generating satisfying user-centric explanations, warnings, workarounds, and the like for users of semi-autonomous driving systems.

**2. Study Examining Social Media Knowledge Sharing Regarding Tesla FSD**

In the following section, we will describe the analysis that we performed on the Tesla FSD social media data.

*2.1. Materials and Methods*

To investigate the kinds of problems and solutions identified in social media by Tesla FSD users, we examined social media posts between the dates of 11 October 2021 and 8 November 2021—roughly the first month immediately preceding and following the broad beta release of the Tesla FSD AI System in October 2021. We used this end date so that we could focus on initial reports and experiences gained within the first few weeks of the system rollout. We examined message boards that specifically enabled threaded conversations, initially identifying 1257 posts related to Tesla FSD. Most of these message boards included many forums or threads that were not about FSD and could be easily eliminated from consideration. From these 1257 initial posts, we identified 101 base posts and 95 threaded responses to posts that directly referred to the FSD system, for a total of 196 total posts. Posts were included if they referred to: an unexpected response or action made by the vehicle; a problem; a safety issue; an illegal maneuver; a negative experience with the decisions made by the vehicle's FSD system; a relevant comment or proffered solution to an issue brought up by another post. We excluded comments that were jokes, memes, off-topic discussions, only praise, comments discussing only safety scores (the method used by Tesla to determine when a driver qualifies to use the FSD mode), software versions, or comments that were Tesla-employee-related.

The 196 resulting posts included 46 from Reddit's "r/teslamotors" group thread for Tesla owners and enthusiasts (https://www.reddit.com/r/teslamotors/, accessed on 11 November 2021), 17 from Facebook's "TESLA Owners Worldwide" group for Tesla owners and enthusiasts (https://www.facebook.com/groups/teslaworldwide, accessed on 11 November 2021), three from Facebook's "Tesla Model 3/Y Owner Technical Support" group for owners and enthusiasts (https://www.facebook.com/groups/teslamodel3 ownertechnicalsupport, accessed on 11 November 2021), three from Facebook's "Tesla Tips & Tricks" group for owners and non-owners (https://www.facebook.com/groups/teslatips, accessed on 11 November 2021), and 127 from online message boards for Tesla owners on AI/Autopilot and autonomous FSD (https://www.teslaownersonline.com/threads/fsdbeta-megathread-for-all-fsd-beta-discussions.18878/, accessed on 14 January 2022). Our complete corpus is available (https://osf.io/6jur3/, accessed on 9 August 2022). We also conducted initial searches on other forums, such as Twitter and StackExchange, but because these did not have a shared forum, the posts were not as systematic, and we excluded these from further analysis.

These 196 posts were used for the communication analysis, which will be described later. These posts frequently included threaded posts involving initial questions and follow-ups, and we included only the follow-up messages that satisfied the criteria that were just described. This corpus of comments was examined in several complementary ways. First, we identified the most common problems appearing in the posts, which we will discuss first. We also examined these from a communications perspective to determine the rationale, communication patterns, reasoning, and motivations for using social media, which will be discussed in a subsequent section.

*2.2. Thematic Analysis of Problems with Tesla's FSD Mode*

To determine the problems experienced by Tesla drivers, we removed 70 posts and responses that we determined were not explicitly about problems experienced by Tesla drivers, leaving 126. We then segmented each comment so that each data record contained only one issue related to the AI system. The 126 total comments related to the AI issues were, therefore, parsed into 273 segments, as some comments addressed more than one issue. Next, each of the segments was summarized by one of the authors, resulting in 119 common summary categories. From those 119 summary categories, 16 codes were

developed via consensus of the authors such that each captured several related summary categories, with the goal that the issues should refer to a similar behavior observed in a similar context, likely stemming from a common part of the automation control system. For example, comments that were initially summarized as "trouble staying in lanes", "turning from a through-lane", and "confused about lane markings" were grouped into a common code of "unexpected lane usage/lane maintenance".

Once these 16 categories were developed, two researchers independently coded each of the 273 statements exclusively into a single category (i.e., a statement could not belong to more than one category). Cohen's $\kappa$ was 0.62, indicating a moderately good level of inter-coder reliability. At this point, the two researchers discussed the differences between their codings, and the coding definition sheet was updated. Finally, the researchers once again independently coded each statement, obtaining a Cohen's $\kappa$ of 0.92, indicating high agreement.

Table 1 summarizes this thematic analysis, and it is organized in rank order from the most to the least common.

**Table 1.** List of coded categories of problems reported by social media users about Tesla FSD.

| Label | Description | Example(s) | Count: Both (Either) Coder |
|---|---|---|---|
| Lane usage | Unexpected lane usage or lane maintenance | Hug center of road/go straight from turn lane | 71 (78) |
| Stopping | Unexpected stopping or slowing down | Phantom braking/stop half a block before the stop sign | 43 (45) |
| Jerky ride | Unnecessary/sudden starts/stops | Jerky turns/brake or accelerate with a sudden jerk | 22 (22) |
| Timidness | Timid Approach | Timid to commit to turn/turn-taking at 4-way stop | 20 (26) |
| Impeding | Impeding other vehicles | Almost impacting another vehicle/following too close | 17 (23) |
| Obstacle speed | Approach impending obstacle too fast or accelerating too fast | Excessive speed at a turn/roundabout | 12 (16) |
| Turning | Improper turning | Wide turns, tight turns, blocking vehicles when turning | 12 (13) |
| Steady speed | Driving too fast/slow for conditions | Unexpectedly driving too fast/slow steadily | 12 (12) |
| Signaling | Improper turn signal usage | Failure to apply, phantom application, wrong turn signal, applies late | 11 (12) |
| Pathfinding | Mismatch between tentacle and actual path | Did not follow GPS route as displayed on screen | 8 (10) |
| Warnings | Inappropriate false system warnings | Inappropriate/false forward collision warnings | 8 (9) |
| Disengagement | Vehicle initiated disengagement or stopped working and did not proceed | FSD stops working/vehicle stops without apparent intent to proceed | 6 (7) |
| Mapping | Unaware of current map configuration | Obsolete/incorrect map data | 4 (6) |
| Camera | Unexpected screen, visualization, camera rendering, or interpretation | Misjudging position of other vehicles/objects | 3 (5) |
| Recognition | Inability to recognize non-road entities | Parking lots, driveways, residential area entrances | 3 (4) |
| U-turns | Problems making U-turns | Avoid/disengage; U-turn turned into a left turn | 2 (2) |

### 2.3. Evaluation of Tesla FSD Problems Identified via Social Media

As demonstrated in Table 1, many of the failures of the system are instances where the human driver's expectations are violated. Indeed, the human driver and FSD putatively share joint objectives (arrive at the destination efficiently and safely while obeying traffic laws), but the operations employed by the human driver and AI FSD to attain that goal differ. Because accidents are rare for both human and AI drivers, it is difficult to argue that a substantial number of these posts were about situations that were, in fact, dangerous. Similarly, although some posts describe situations such as failures to signal that might result in traffic violations, most relate to behaviors that are within the bounds of legal vehicle operation but violate expectations of human drivers. For example, we suspect that the most common complaint—improper lane usage—was rarely an actual safety issue or

traffic violation. On unmarked city streets in right-side-driving countries, human drivers will naturally stay on the right side of the road even when there is no traffic, and this is likely done out of adherence to driving habits and the anticipation of exceedingly rare events. However, even in this case, the users would describe taking over control in order to avoid a potential collision, so some safety risks clearly existed.

If an emergent cooperative intelligence between the human and FSD occurs, the human ultimately remains in a supervisory role. Thus, the human must maintain vigilant awareness of the current state and possible future states and must provide input when necessary. Our examination of social media posts suggests that default cognitive anthropomorphic assumptions are often violated, such that the FSD operates in ways that differ from the expectations of experienced human drivers. Understanding these issues will allow drivers to anticipate them, avoid them, and develop workarounds to mitigate any safety risks involved.

The problems that we identified via Tesla social media have a broad scope—from minor annoyances to major issues that could cause accidents and casualties. Some of these problems are reported much more frequently than others (e.g., lane usage vs. mapping problems), and we might infer that the number of reports is roughly related to the number of occurrences by drivers, with a few caveats. First, we suspect that the likelihood of reporting will also depend on the seriousness of the problems. For example, minor issues that happen very frequently may go under-reported. On the other hand, rare problems may be over-reported because they might be viewed anomalous and, thus, trigger a user's willingness to investigate further. Similarly, very serious problems leading to accidents and injuries (although they are likely to be rare) might go unreported if legal or insurance remedies are being pursued.

In summary, our evaluation of social media posts suggests that it can indeed be the source of collaborative explanations for AI automation. We believe that it might also serve as a starting point for training and tutorials, an idea that we will develop more in the general discussion. First, we will report an examination of the communication patterns exhibited in the social media posts to understand the kinds of explanations that are provided, the motivations for posting, and the general suitability of this kind of social media for supporting users of AI.

*2.4. Examination of the Kinds of Expectation Violations Identified in the Corpus*

Most of the problems reported constitute a mismatch between user expectations and the system behavior. However, there are several different ways in which the behaviors differ from our expectations, which we will classify as related to *safety*, *legal issues*, *interface and interaction*, and *cognitive anthropomorphism*. First (safety), we expect the system to operate safely and not get into an accident. Second (legal), we expect the system to follow the rules of the road and not violate regulations, speed limits, etc. Third (interface), we expect the interface (HUD, GPS route, warnings) to match the behavior of the vehicle and the actual environment. Finally, (cognitive anthropomorphism), we expect it to drive according to the norms and typical behaviors of other human drivers—even if this does not break a law or result in a dangerous situation.

First, vehicles should not perform dangerous maneuvers and should transport us safely to our destinations. We saw a number of comments that illustrate this kind of violation in the corpus. For example, one comment reads "…it will sometimes try and pull out in front of approaching vehicles", and another states "Still has the bad habit of wanting to go into oncoming traffic to get around vehicles in stop and go traffic. Even when it senses brake lights." These comments illustrate how some of the Tesla FSD behaviors are perceived as dangerous or unsafe. Making users aware of these situations can be important in order to allow them to anticipate the need for takeovers, although a better solution would be to improve the autonomous system for it to operate more safely.

Second, drivers expect vehicles to follow the rules of the road and other laws. A number of comments were in response to violations of this kind of expectation. For example, in one case, the driver reported an event that "…happened at a red light with

a car in front of me. When the light turned green, my car attempted to pass the car in front using the left turn lane. It was pretty easy to tell this might happen as the visualized predicted path of my car kept jumping back and forth between following the car in front and going around it". Another driver stated that the turn signal usage did not follow the rules of the road: "It put them on multiple times when the road just goes around a bend, or at a four way stop when going straight ahead at just a slight angle. Really confusing to other drivers who thought I was going to turn onto a different street". These behaviors may not always be dangerous, but they do represent rule-breaking that is unexpected and warrants reporting to the community message boards.

Third, drivers should expect the parts of the multi-modal system of sensors, warnings, displays, and controls to be consistent with one another and the environment. An example of this inconsistency is "The vehicle doesn't make the necessary turn even though the navigation shows the turn is needed", and "in most cases the tentacle has a pretty accurate representation of what the car should do. Unfortunately it doesn't always follow the tentacle." Many of these expectation violations involve errors in the software system or interface choices that need to be fixed by designers, but social media may help users understand that it is a systematic problem and not just an issue with their own vehicle or their understanding of the vehicle.

Finally, drivers may expect vehicles to behave as a human driver would [17], and we might expect this is their default expectation, which we have referred to as *cognitive anthropomorphism* [2]. Many of the reports fall into this category—for example, one driver stated that "on interstates, the car wants to change lanes in front of a car that is rapidly approaching on the left." Another driver thought an unprotected left turn impinged on another, "too tight of an unprotected left turn, crowding other car turning right". These situations may not be illegal, incorrect, or even unsafe, but represent driving behavior unlike that of human drivers. These are cases where educating and informing drivers based on the experiences of others may be especially helpful.

This examination shows that posts about the FSD mode often appear to be motivated by a violation of expectations. These violations fall into a number of categories, from safety and legal aspects to expectations for the interface and those of human-like driving behavior. In many of these cases, even if the best course of action is for developers to improve the system, knowing about these problems may help a user understand, anticipate, and avoid the problems that might ensue.

## 3. Examining Social Media Comments as a Collaborative Support System

The collaborative user-driven help embodied in the message boards that we examined provides much of the functionality proposed in the Collaborative XAI (CXAI) system [3], which was modeled on social Q&A (SQA) platforms with specific elements to support the explanation of AI systems. Many SQA platforms focus on bug fixing (such as StackOverflow or StackExchange), and they can be associated with 'how-to', 'why', and 'why-not' explanations [25]. An earlier evaluation of the observations from the CXAI system showed that a considerable proportion of posts involve describing 'What', with a smaller but substantial proportion answering 'Why' questions [26]. 'What'-style posts describe problems and limitations, but do not offer fixes or workarounds. In the case of commercial AI systems such as FSD, this may be ideal because the general user will not be involved in fixing problems, but being aware of the problems may help them operate more safely.

Nevertheless, our past examination of posts in a sample CXAI system showed that these were missing a considerable number of what might be called reasoning traces [27]—logical discourses providing causal explanations of a system's behavior. So, it may be beneficial to know what types of communication occur in social media related to semi-autonomous vehicles in order to determine (1) what kinds of communication are supported and (2) if more structured SQA features might encourage additional styles of communication and explanation. This will help to understand how these social media systems can help support cooperative intelligence between drivers and the autonomous systems that they are supervising.

Consequently, we examined the Tesla social media communications from a number of complementary perspectives to give a better understanding of their goals and purposes. This involved four complementary coding dimensions. These are *reframing*, *resolution*, *emotion*, and *cognitive empathy*.

### 3.1. Coding Schemas

Each response was coded according to four independent dimensions, each with several coding labels. These include:

1.  Reframing: The way in which a statement updates or modifies a user's thinking about the AI system. This coding scheme was developed based on research on team communication [28,29].
2.  Resolution: Whether the problem was resolved. We included this coding to determine how many of the posts resulted in satisfactory answers to queries, which Mamun et al. [26] determined occurred relatively rarely in CXAI systems.
3.  Emotion: The statement that showed emotions such as frustration, etc. [30].
4.  Cognitive Empathy: Empathetic elements such as sharing an experience, understanding someone's feelings, or sharing information [31].

A more detailed description of the coding criteria are shown in Table 2.

**Table 2.** Coding schemas for communication about AI systems.

| Dimensions | Elements/Labels | Description/Clarification |
| --- | --- | --- |
| Reframing | Evaluation | Evaluative utterances or judgments concerning the activities of the scenario just played out. Analyses of why things went well or wrong. |
| | Clarification | Questions and answers that someone either asked or seemed to misunderstand. This includes repetitions for clarification, associations, and explanations. Clarifications serve to clear up misunderstandings from other individuals. |
| | Observation | A statement that describes the AI's action during use. |
| | Response uncertainty | Statements indicating uncertainty or lack of information with which to respond to a command, inquiry, or observation. |
| | Denial or Disconfirmation | Disconfirming a statement. |
| Resolution | Situation Resolved | Combination of some of the other elements of Reframing—resolution/ workaround/abandonment of a practice conditionally/abandonment of a practice wholly/why it is doing it (not giving a solution but a reason). |
| Emotion | Frustration or anger with AI | During the use of AI. |
| | Frustration or anger on response | During the use of AI. |
| | Appreciation for the AI | During communication. |
| | Appreciation for a Response | During communication. |
| | Embarrassment | Any response apologizing for an incorrect response, etc. |
| Empathy | Agreement/Acknowledgement | 'A' conveys to 'B' that the expressed emotion, progress, or challenge is legitimate. |
| | Shared experience | 'A' has a similar experience to that of 'B' with progress or a challenge. |
| | Perfunctory recognition | 'A' gives an automatic, scripted-type response, or repeats company's policy/response, giving the empathetic opportunity minimal recognition. |
| | Antagonism | Deflates the other's response, defends or asserts self-response. |

### 3.2. Method

Two coders independently coded 196 observations (posts and comments) regarding AP and FSD on each of the four dimensions. The coding was dependent on the context, so the coding of a single comment was dependent on the parent post and earlier comments. If a comment was deemed to be a separate post based on its uniqueness and child comments, the coders separated it as a new observation after coming to a consensus. Following an initial round of coding on a subset of items, the coders met to examine disagreements in coding, then completed a second round of coding.

### 3.3. Results

The results of this coding are shown in Table 3. As with Table 1, we provided counts of statements for which both coders agreed and counts for statements for which at least one coder specified the category.

**Table 3.** Results of the communication coding for each dimension within each coding scheme. The counts shown are the numbers that both coders agreed on and the numbers that at least one coded for each dimension.

| Dimensions | Elements/Labels | Both (Either) Coder |
|---|---|---|
| Reframing | Evaluation | 20 (34) |
| | Clarification | 23 (31) |
| | Response uncertainty | 16 (19) |
| | Observation | 111 (135) |
| | Denial or Disconfirmation | 1 (3) |
| | Other | 1 (8) |
| Resolution | Situation Resolved | 38 (60) |
| | Not resolved | 136 (158) |
| Emotion | Frustration or anger with AI | 96 (120) |
| | Frustration or anger on response | 0 (2) |
| | Appreciation for the AI | 27 (36) |
| | Appreciation for a Response | 0 (3) |
| | Embarrassment | 0 (1) |
| | Non-emotional | 47 (60) |
| Cognitive Empathy | Agreement/Acknowledgement | 23 (54) |
| | Shared experience | 21 (33) |
| | Perfunctory recognition | 0 (1) |
| | Antagonism | 9 (12) |
| | Non-empathy | 124 (138) |

#### 3.3.1. Reframing

For Reframing, the coders achieved a strong agreement, with $\kappa = 0.78$ [32]. More than half of the comments were coded as 'observations', consistent with our previous work. It is useful to acknowledge that this kind of explanation system will not generally provide answers to 'why' questions, in part because the answers are often not knowable by general users and will be dependent on the context of the situation, which might not be conveyed in the online format. However, the 'what'-style posts are important, and they offer a kind of information that traditional XAI systems miss. Some example statements coded in each major category of reframing include:

- *Reframing: Evaluation.* A user summarized other reports about FSD turning behavior in comparison with that of AP (Record #12-1364).
- *Reframing: Clarification.* Commenter asked which driving profile (chill, average, or assertive) was set (Record #77-1490).
- *Reframing: Response uncertainty.* User asked whether only a subset of adaptive cruise control features could be used (Record #14-1371).
- *Reframing: Observation.* User observed some conditions under which FSD was not working (Record #1-1338).

#### 3.3.2. Resolution

For the Resolution dimension, the agreement was moderate ($\kappa = 0.7$), with around 20–30% involving resolution. The majority of the posts and comments in the Tesla communication chain are regarding the AI's action during use. However, this is informative because most resolutions are limited to responses to other comments, and they often mark the end of a discussion thread. An example of a statement coded as a resolution follows:

- *Situation resolved.* User suggests turning off sentry mode to avoid conflict between FSD and AP (Record #2-1343).

### 3.3.3. Emotion

The posts mostly showed frustration or anger with the AI, with very little appreciation for the system (moderate agreement, $\kappa = 0.78$). It may not be surprising that reports about failures of the system are associated with anger and frustration, but it is likely that this emotional reaction helps to motivate users to post their experiences, which will benefit other users indirectly. Some examples of statements coded in each emotion category include:

- *Emotion: Frustration/Anger.* User 'yells' "so ALWAYS be prepared to take over" (Record #8-1355).
- *Emotion: Appreciation.* User states "...I've enjoyed it so far" (Record #10-1360).

### 3.3.4. Cognitive Empathy

The users showed a tendency to help out other users by offering advice/support or elaborating on a situation. This shows cognitive empathy [31]; however, this occurred for only a relatively small subset of comments (approximately 20%). The coders had a strong agreement ($\kappa = 0.81$) for empathy. Some example statements include:

- *Cognitive Empathy: Agreement.* User agrees by saying "Yes!" to a previous post about the future of AP navigation (Record #52-1449).
- *Cognitive Empathy: Shared experience.* User shares a similar experience of their first drive using FSD (Record #28-1410).
- *Cognitive Empathy: Antagonism.* User said that a previous user's account about turning "...was NOT my experience" (Record #31-1415).

### 3.4. Discussion

The analysis showed that user communication focusing on understanding a new system mostly (up to 75%) involves 'what' type of observation–explanations. Despite this, posts showed a tendency to support resolution (up to 30.6% of statements). However, many of the resolutions remained perfunctory, and specific motivations [26] may be needed to encourage resolutions with more complete reasoning traces.

Several of the codes highlight how these discussion forums indeed support dialectic and discourse. In general, clarification posts indicate a response to requests for more information; resolution posts represent the end of threads that provide an answer or workaround to the initial query; agreement/acknowledgement, shared experience, and even antagonistic posts involve a thread of discussion or discourse. This is unlike many bug-reporting systems, which involve more asymmetric discourse (a QA team member may request more information or mark a bug as 'won't-fix', but discussion and 'me-too' responses are often discouraged), and it is even different from SQA systems such as StackExchange, which are more transactional, providing upvotes for correct answers as judged by the user base.

The coding for emotional content is interesting because it shows that up to half of the comments had demonstrations of emotion—and mainly negative emotion. Emotional state is motivational, and this suggests that a major reason for people visiting and reporting problems is their own frustration and anger. Within the SQA research domain, extensive research has been conducted to understand motivations for using the system, but they mainly focus on either aspects, such as accuracy/completeness/timeliness of responses or gamification features, such as reputation, badges, and upvoting to encourage participation. This suggests that emotion may be a powerful intrinsic motivator for participation.

Empathetic elements, such as showing shared experiences and agreement, are present in a small but substantial subset of the statements. The statements showed empathetic collaboration between users, which one might not expect from a bug-reporting system hosted by the system vendor. This suggests that users do engage in the social platform based on a shared interest with a cooperative group identity, which provides another intrinsic motivation—something one might not expect in reports to government safety boards or vendor-sponsored bug-reporting systems.

Overall, this analysis suggests both strengths and limitations of social media and SQA systems for supporting user understanding of an AI system, which, in turn, can help support the cooperative cognition involved in supervisory control of semi-autonomous vehicles. The clear strength of the system is that it helps identify possible problems with the automation, especially the contexts in which the failures happen. This, in turn, can help users anticipate and be ready to take over or to avoid those situations altogether. Additionally, our coding suggests that aspects of social media encourage reporting, dialog, and clarification. This is partly fueled by frustration and anger that drivers experience (which may tend to make them avoid official bug reporting), but also shows evidence for cooperative and empathetic interactions that encourage helping one another. Supporting this community spirit, we suspect that the majority of the benefit is derived from readers of the discussion, rather than the posters themselves, as it illustrates situations that they may experience in the future and warns them about the system's operation.

Some limitations include the possible biases in reporting, the timeliness of these reports for systems that are constantly improving, the possibility that users are mistaken or incorrect about their reports or their explanations, and the unstructured and ad hoc nature of distributed social media. Some of these limitations might be avoided by developing a more formal and structured SQA system, such as the CXAI concept [26]. In such a system, additional structure might provide better verification by experts or officials, easier methods for showing agreement, and other schemes to encourage broader participation. Another route is to use the problems and lessons uncovered by such a system to develop tutorials that more explicitly teach users about the limitations of the system—a concept that will be explored in the General Discussion.

## 4. General Discussion

We will now discuss cooperative cognition and expectations of an AI system, as well as concepts for using social media posts as cognitive tutorial training material.

### 4.1. Cooperative Cognition and Expectations of AI Systems

The cooperative cognitive interaction between an autonomous vehicle and a driver is asymmetric. Except for minor parameter or mode settings (e.g., how much the vehicle is allowed to exceed the speed limit, etc.), the AI system does not adjust its behavior to meet the expectations of the user. Even for parameter setting, it is the user who actively adjusts the AI system. There are in-vehicle displays that help the system show the user what it is "thinking" or "seeing", but the burden remains on the user to adapt to the system's limitations. The main ways in which a user can control the system are via (1) limited parameter settings, (2) anticipation and avoidance of situations that might cause problems, and (3) careful monitoring and takeover in situations that appear dangerous, which are often situations that violate the user's expectations of how a human driver should operate. We have found that each of these control modes is supported by social media posts, which, although similar to vendor bug databases and government safety board reports, can serve a very different purpose. Their goal is to report to and seek feedback from a community, rather than an authority, and they allow the rest of the community to benefit from discoveries made by a user. This helps users understand whether a particular problem with a rare occurrence is likely to happen in other situations. This also helps users clarify and understand the problem with feedback from others. Fundamentally, it helps users develop a better mental model of the AI system, which can help support appropriate reliance and trust—knowing when the system should be used and when it should be avoided. Thus, the main benefit of these community-based help systems is that they allow users to develop a better understanding of the system in order to anticipate and avoid problems.

### 4.2. Concepts for Using Social Media Posts as Cognitive Tutorial Training Material

One of the limitations of social media for supporting explanations of AI systems is their distributed and ad hoc nature. For example, we identified about 200 posts out of more than 1200 relevant posts about FSD, and this ignores the thousands of Tesla-related posts in the time frame that were not about FSD at all. Furthermore, we obtained these from five different forums (not including Twitter or traditional SQA systems), and users are unlikely to be aware of or tracking all of them.

This suggests the potential for focused tutorials that summarize the common problems with an AI system and train users to recognize and identify these problems. We suggest that AI systems can be supported by *Cognitive Tutorials*, experiential and example-based training approaches that focus on the cognitively challenging aspects of an AI system. Previously, we described a variety of approaches for generating these tutorials [33–35], including how social media can be used to develop support their development. Our previous cognitive tutorials were developed from a variety of sources, but we suggest that SQA posts may form an ideal corpus on which to identify high-priority learning objectives, identify cases and situations that can be used for training, and generate reasonable resolutions or warnings that enable users to anticipate and avoid problems.

One cognitive tutorial format that we have been evaluating in our lab involves what we call "explicit rule learning". The goal of this kind of tutorial is to identify a probabilistic relationship between conditions and the behavior of an AI system—usually in terms of proper and improper performance. This contrasts with explanation approaches that help a user learn about a relationship implicitly via multiple examples or feedback. For an AI classifier system, the implicit approach might be to show many examples of correctly and incorrectly identified images (e.g., of dogs) and allow the user to infer the kinds of images that will be erroneously classified (e.g., noticing that the AI makes errors when the dog is with a human, or in the outdoors, or for certain breeds). In contrast, our explicit rule learning approach provides a "rule card" that gives an at-a-glance electronic or paper description of a probabilistic rule in four parts: an explanation of the rule, visual examples, rule sensitivity, and a summary of the rule's effectiveness/base rate. Each rule card contains one issue and might show times when the system fails, systems' actions that violate our expectations, or adverse actions by the FSD that are likely to occur.

As an illustration, Figure 1 shows an example rule card that we developed to train one of the most common complaints of the Tesla FSD: lane-keeping. In the corpus, a number of social media posts were summarized in such ways as "Hogs the road if not striped in center", "still on right side but leans to center", and "moves over for traffic." This rule card begins with a statement of the problem and the rule. Next, we show examples (via in-vehicle images) that help illustrate the situation. These might alternately be video or top-down illustrations if these are available. The goal of this section is to provide a variety of examples that show varied conditions in which the problem occurs. Next, we show a thermometer that helps illustrate the likelihood of the rule being accurate. Estimating the likelihood directly is difficult via social media, so these are generated from estimates of the tutorial developers; we also developed such estimates directly from other analytic data when available. Finally, the last section gives a statement about its incidence, which can help the user understand how often the situation might occur and, thus, determine whether it is relevant to their own usage.

We are currently in the process of testing the effectiveness of these tutorials on users, but have found success in several unpublished studies. We believe that the basic format can provide an alternative approach to leveraging social media data in order to help users of AI systems understand and predict their limitations.

**Description:** When FSD encounters a two-lane residential road without road markings, it may stay to the right as expected, or it may tend to drive down the middle of the road, centered in both lanes. You can see examples below. Not having road markings in a two-lane residential street confuses the AI so that it often errs by driving in the center of the lanes, even in the path of oncoming vehicles

Example two-lane road **with** lane markings

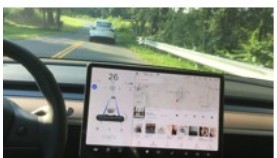 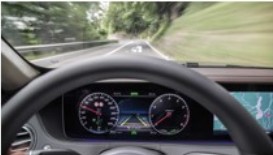

Usually drives on the right side as expected

Example two-lane road **without** lane markings

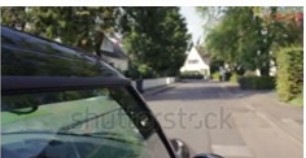 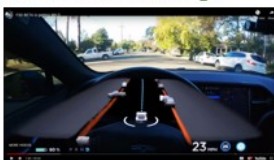

Usually drives in the middle of the road, centered in both lanes

Out of all cases **with** lane markings, most vehicles drove on the right side as expected

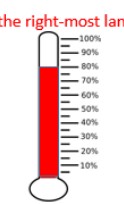

Out of all cases **without** lane markings, most vehicles drove in the middle of the road, centered in both lanes

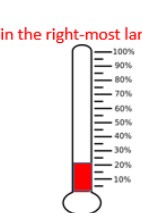

**Importance:** This rule is applicable mainly for residential streets, which is a frequent situation for many drivers. Reports suggest that the vehicle frequently behaves this way on residential streets without road markings, and rarely does so on non-residential streets or residential streets without road markings.

**Figure 1.** Illustration of a Cognitive Tutorial rule card to help support learning a systematic limitation of the Tesla FSD mode.

*4.3. Limitations*

Previously, Mamun et al. (2021a, 2021b) [3,26] examined how a collaborative SQA platform (CXAI) could be used to help users understand the limitations of an AI system. However, those studies used a special-purpose platform for a simple image classifier that was used only for evaluating the CXAI system. This study represents our attempt to demonstrate that existing social media platforms may provide the same kinds of information for existing commercial AI systems, and we selected Tesla's FSD system as a target because it is one of the first complex AI systems that has gained wide consumer exposure. In order for this to be maximally effective, social media posts about the system need to be representative, comprehensive, and unbiased. However, it is difficult to determine whether this is the case. First, reporting may be biased in a number of ways—over-reporting some problems and under-reporting others. Second, online review systems are often manipulated by bots or malicious actors, and because problems reported via these social media platforms are not verified, it is possible that some proportion of the posts are fabricated (maybe by activists

who oppose Tesla, competitors, or investors trying to manipulate stock prices). Third, reports may be biased, in that only some kinds of users actually make posts about problems, only some kinds of problems will get reported, users may be mistaken about the details or causes of the problems they actually report, and the reports are sometimes difficult to interpret. Thus, the reports may be non-representative and inaccurate. However, we believe that the number of erroneous or malicious posts is small, as many of the reported limitations—especially the most common ones—were discussed and confirmed by others in follow-ups. Furthermore, our coding by multiple raters demonstrated that these reports are reasonably systematic. We do think that the risk of bias may increase if social media posts become an important source for regulatory oversight or comparisons between systems because the consequences of a bad or malicious post may be greater.

Although this demonstration was focused on a snapshot early in the deployment of Tesla FSD for a good reason (it is the most widely deployed semi-autonomous system in on-the-road vehicles), it also represents a potential limitation—many of the problems identified might not occur in other semi-autonomous vehicles or in later versions of the FSD system. Nevertheless, the list of problems that we identified demonstrates how AI systems that replace a task that humans routinely carry out can violate our expectations because they are unsafe, illegal, or just differ from the behavior of human operators. We believe that some of these behaviors are likely to occur in other semi-autonomous systems, and this may provide a road map for developers of future self-driving systems.

*4.4. Training as a Means of Improving Cooperative Cognition in Multi-Modal Technology Interactions*

There are many ways in which designers can help support human users of complex multi-modal technologies and AI systems. These often focus on human-centered design principles [36] that help explain the AI to users, make it transparent, enable more direct control, and improve situational awareness. However, the goal of these systems is often to help the user understand how the system works. To the extent that XAI or transparency enables the development of a better mental model of the system that one is using, this helps anticipate and avoid problems, as well as potentially address problems after they have happened. For example, vehicles will often show a display of the vehicles and other obstacles that it detects in order to inform the user when there are potential hazards it is not aware of. However, these approaches are often retrospectively helpful in local situations—explaining why a decision was made or showing a warning when it enters an error mode. These can be helpful in understanding a specific situation and may help in developing a better long-term understanding of a system, but we suggest that more direct scenario-based training may be more effective and efficient [33,34]. Thus, for systems such as FSD that are unlikely to adapt or cooperate with a user in meaningful ways in the near future, the best approach is to help the user adapt and cooperate with the system.

In other contexts, we explored a number of ways to develop such training about AI systems [35]. For example, systematic user tests, cognitive task analysis interviews, discussions with system developers and designers, a review of documents and artifacts, and a deep analysis of algorithms are all complementary approaches to identifying learning objectives that can help a user. The approach that we described in this paper outlines a systematic way of identifying learning objectives for training and tutorials from social media and represents a relatively low-cost way of approaching this, provided that there are enough users and common social media forums or groups where they discuss their use of the system. In fact, encouraging and supporting user-to-user discussion and problem reporting is likely to be a useful way of identifying the limitations and either fixing them or creating training for future users.

**5. Summary and Conclusions**

In this paper, we evaluated social media posts appearing in the weeks following the release of the Tesla beta FSD semi-autonomous driving mode. We suggest that this

information can provide users with helpful information that supports cooperative cognition in supervisory driving modes, especially in terms of their ability to recognize potentially dangerous situations, anticipate problems, avoid them, and be faster at taking over control of the vehicle when they occur. The posts cover a number of different problems experienced with vehicles and appear to primarily cover four categories of expectation violation: safety, legal, interaction, and anthropomorphic aspects. Although the best courses of action for remedying these situations differ for each category, awareness of these problems can always help a user anticipate and avoid situations that might lead to the problems.

Along with a comprehensive analysis of the kinds of problems that FSD drivers experience, we also examined the communication patterns in social media to understand how they support the creation of useful accounts of the limitations. Finally, we provided an example tutorial format that might be helpful in addressing some of the weaknesses identified in social media and might enable users to learn more directly about the problems and limitations of the AI systems that they are using. Together, these suggest that social media can provide a useful resource for users of AI, automation, and autonomous vehicles in particular. Importantly, the resource helps users develop more accurate mental models of the system's behavior, enabling better anticipation of problems and the ability to enhance the cooperative cognition in which the user–AI system is engaged.

**Author Contributions:** Conceptualization, methodology, and writing, A.L., T.I.M. and S.T.M.; data collection and coding, A.L. and T.I.M. All authors have read and agreed to the published version of the manuscript.

**Funding:** This research received no external funding.

**Institutional Review Board Statement:** Not applicable.

**Informed Consent Statement:** Not applicable

**Data Availability Statement:** The complete corpus of social media posts is available at https://osf.io/6jur3/, accessed on 9 August 2022.

**Conflicts of Interest:** The authors declare no conflict of interest.

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
