# Peer review of "When Self-Driving Fails: Evaluating Social Media Posts Regarding Problems and Misconceptions about Tesla’s FSD Mode"

_mti, doi:10.3390/mti6100086_

Round 1

Reviewer 1 Report

Reviewer's summary after reading the manuscript:

Consumers are using semi-autonomous vehicles for both highway and residential driving for the first time with the recent deployment of the next generation of Tesla's Full Self Driving mode (FSD). As a consequence of this, drivers are forced to deal with difficult and unexpected circumstances while utilizing untested technology, which presents a significant obstacle to cooperative cognition. Informing and educating the user on the potential limitations of the scenario is one approach that may be taken to enhance cooperative cognition in settings like this. Users have turned to the internet and social media in order to document their experiences, seek answers to questions they have, provide advice on features to others, and assist other drivers who have less FSD experience. This is because it is not easy to learn about the limitations associated with these devices. In this research, the authors investigate how posts on social media might assist identify the limitations of automation, as well as how this can provide collaborative, crowdsourced support while also being beneficial for tutorials and other forms of training. In this paper, the authors examine a corpus of posts on social media about FSD problems in order to examine (1) the typical problems reported; (2) the types of explanations or answers provided by users; and (3) the practicability of utilizing such user-generated information in order to provide training and assistance for new drivers. The results reveal a number of limitations of the FSD system (such as phantom braking and lane-keeping), which may be anticipated by drivers. Having access to this information would enable drivers to predict and avoid the problems, which allows for better mental models of the system and supports cooperative cognition of the human and AI system in more situations.

----------------------------------------

Dear authors, thank you for your manuscript. I enjoyed reading it. Presented are some suggestions to improve it:

(1) Please consider modifying the title of the manuscript to include the words "cooperative cognition" since your paper is centered around the core theme of "cooperative cognition" so that it would be easier for potential readers to find your study. 

(2) Please consider including "cooperative cognition" as one of the keywords.

(3) Please include a "Limitations of This Study" section to discuss what were the challenges faced, and how your team overcame those challenges. This would be very beneficial to the readers as they would be able to learn from your expert knowledge.

(4) To improve the impact and readership of your manuscript, the authors need to clearly articulate in the Abstract and in the Introduction sections about the uniqueness or novelty of this article, and why or how it is different from other similar articles. Can the authors please kindly elaborate more about how this study is relevant to the improvement and usage of "multimodal technologies and interaction" since it was submitted for publication in the journal entitled "Multimodal Technologies and Interaction"?

(5) Please substantially expand your review work, and cite more of the journal papers published by MDPI.

(6) All of the references cited are not yet properly formatted according to MDPI's guidelines. For example, the DOIs of all the journal papers cited are not included yet. For the references, instead of formatting "by-hand", please kindly consider using the free Zotero software (https://fanyv88.com:443/https/www.zotero.org/), and select "Multidisciplinary Digital Publishing Institute" as the citation format, since there are currently 32 citations in your manuscript, and there may probably be more once you have revised the manuscript.

Thank you.

Author Response

Sep 9, 2022

 We appreciate your time and valuable feedback on our paper, thank you. It’s a great opportunity for us to improve this paper. We’ve considered all of your suggestions, and we believe it has indeed made the paper better. Most critically, we’ve clarified our Method section, added a Limitations section, and added several sections that deepened our analysis and conclusions about our corpus.  We have also added content that makes this contribution more germain to multimodal technology and cooperative intelligence.

We do have several questions. First, the link to the manuscript under “Please download the latest version of the manuscript for revision. Your original submission may have been changed.” Manuscript for Revisions takes us to a copy of the manuscript from July 12, 2022, well before our final submission. However, if we download the file “manuscript.pdf”, it seems correct (July 22, 2022 version).  We have updated the most recent version, but were puzzled about how this previous version got into the system.

Secondly, several reviewers made note of our Reference formatting. We used the BibTex (Overleaf) template, and we’re not sure how to change the predetermined formatting. Any assistance from the editors with this would be greatly appreciated.

Below is a list of each comment and how we addressed them.

Comment

1: (1) Please consider modifying the title of the  manuscript to include the words "cooperative 

cognition" since your paper is centered around  the core theme of "cooperative cognition" so that it would be easier for potential readers to find  your study.  

Response

We have considered this, but given it appears in a special issue on cooperative cognition we are not sure if this is necessary.

Comment:

1: (2) Please consider including "cooperative cognition" as one of the keywords.

Response:

 We have added this as a keyword

1: (3) Please include a "Limitations of This Study"  section to discuss what were the challenges 

faced, and how your team overcame those  challenges. This would be very beneficial to the  readers as they would be able to learn from your  expert knowledge.

Response:

We have added a limitations section focusing primarily on the reliability and biases of social media data for understanding limitations of AI systems.

Comment:

1: (4) To improve the impact and readership  of your manuscript, the authors need to clearly articulate in the Abstract and in the  Introduction sections about the uniqueness  or novelty of this article, and why or how it  is different from other similar articles. Can 

the authors please kindly elaborate more  about how this study is relevant to the 

improvement and usage of "multimodal  technologies and interaction" since it was 

submitted for publication in the journal entitled "Multimodal Technologies and Interaction"?

Response:

We have edited the intro and abstract to make a stronger connection to MTI and to highlight the novelty of our approach. However, our paper is focused on the notion of cooperative cognition, as identified by the special issue “cooperative intelligence and automated driving”; and trying to link to the mission of the journal and the alternate mission of the special issue ends up adding substantial inconsistent goals of the paper, so our arguments all focus on improving MTI via cooperative intelligence.

Comment:

1: (5) Please substantially expand your  review work, and cite more of the journal 

papers published by MDPI.

Response:

We have expanded our work reviewing related papers published by MDPI.  It is hard to respond to a non-specific request to expand the review work—are we missing relevant research that has used social media to evaluate AI or semi-autonomous vehicles? We don’t think so. We disagree that a comprehensive review of research on self-driving vehicles is necessary for this paper, and would not change our findings or our interpretation. 

Comment:

1: (6) All of the references cited are not  yet properly formatted according to MDPI's 

guidelines. For example, the DOIs of all  the journal papers cited are not included 

yet. For the references, instead of  formatting "by-hand", please kindly 

consider using the free Zotero software  (https://fanyv88.com:443/https/www.zotero.org/), and select

"Multidisciplinary Digital Publishing Institute"  as the citation format, since there are 

currently 32 citations in your manuscript,  and there may probably be more once you 

have revised the manuscript.

Response:

References were not formatted by hand, but using zotero and bibtex in latex. The MDPI bibtex settings are automatically set via the journal’s preferred bibtex style file, which itself is a bit buggy and cannot handle some very common citation formats like web pages.  We have updated dois for articles where they are available, 

Reviewer 2 Report

Regarding the content, I do not have any changes to recommend, it makes a good literary review to support the relevance of the problem to be studied and a good structuring of the content, it uses the correct methodology for this type of study and it is a consistent and well-detailed methodology to give significance to the results they show, makes a good discussion of the results with respect to the studies carried out previously, and marks the conclusion obtained well.

Although I advise looking at these things:

Respect the template provided by the journal, the first page does not have the journal logo.

In the information of the authors, you must add the email of the authors. Look at the template.

Never two sections without a paragraph of text in between. You should put a couple of lines describing/naming the subsections you are going to deal with within that section. You must correct this between sections 2-2.1 and 4-4.1.

In the section “5. Summary and Conclusion”, it is necessary to develop a deeper analysis of the conclusions, implications and limitations of the study. In addition to the possible future lines of research opened with this research.

And the references in the 'References' section must follow the model set by the journal. You must correct the errors that exist. Look at this in the template.

Author Response

Sep 9, 2022

To the Editor:

 We appreciate your time and valuable feedback on our paper, thank you. It’s a great opportunity for us to improve this paper. We’ve considered all of your suggestions, and we believe it has indeed made the paper better. Most critically, we’ve clarified our Method section, added a Limitations section, and added several sections that deepened our analysis and conclusions about our corpus.  We have also added content that makes this contribution more germain to multimodal technology and cooperative intelligence.

We do have several questions. First, the link to the manuscript under “Please download the latest version of the manuscript for revision. Your original submission may have been changed.” Manuscript for Revisions takes us to a copy of the manuscript from July 12, 2022, well before our final submission. However, if we download the file “manuscript.pdf”, it seems correct (July 22, 2022 version).  We have updated the most recent version, but were puzzled about how this previous version got into the system.

Secondly, several reviewers made note of our Reference formatting. We used the BibTex (Overleaf) template, and we’re not sure how to change the predetermined formatting. Any assistance with this would be greatly appreciated.

Below is a list of each comment and how we addressed them.

Comment:

2: Respect the template provided by the journal, the first page does not have 

the journal logo.

Response: This is built into the latex template and we did not change anything.

Comment:

2: In the information of the authors, you must add the email of the authors. 

Look at the template.

Response:

We have updated this.

Comment:

2: Never two sections without a paragraph of text in between.  You should put a couple of lines describing/naming the subsections  you are going to deal with within that section. You must correct this  between sections 2-2.1 and 4-4.1.

This rule stated by the reviewer is non-standard,  publication-specific, and does not appear in MDPI author layout style guide, or in another paper published in this special issue https://fanyv88.com:443/https/www.mdpi.com/2414-4088/6/9/73

Comment:

2: In the section “5. Summary and  Conclusion”, it is necessary to develop  a deeper analysis of the conclusions,  implications and limitations of the study.  In addition to the possible future lines  of research opened with this research.

Response:

We have incorporated additional analysis in the text–especially in a section focusing on how many of the posts involve different forms of expectation violation. We have updated the conclusions section to reflect this, but have not made extensive additional edits. We do not see a rhetorical value in suggesting future lines of research.

Comment:

2: And the references in the 'References'  section must follow the model set by 

the journal. You must correct the errors  that exist. Look at this in the template.

Response:

We are using the mdpi bibtex template, which doesn’t handle proceedings, or web pages, or reports according to mpdi format.It is unclear what we can do about this without assistance from the editorial office.

Reviewer 3 Report

I want to thank the opportunity to review the manuscript titled " When self-driving fails: Evaluating social media posts regarding problems and misconceptions about Tesla FSD mode". I also want to thank the author(s)'s efforts in conducting this study. This study investigates a hot and interesting topic—self-driving. This paper is well-written and easy to read. The result of the study is meaningful and useful for Tesla.

However, due to the study design, this study made very little contribution to the body of knowledge in the field of self-driving. First, this study is highly descriptive. There is no theoretical contribution. Second, the context of this study is Tesla self-driving. The results of this study may not be applied to other self-driving systems. For these reasons, this study cannot be seen as a research paper. 

Author Response

Sep 9, 2022

To the Editor:

 We appreciate your time and valuable feedback on our paper, thank you. It’s a great opportunity for us to improve this paper. We’ve considered all of your suggestions, and we believe it has indeed made the paper better. Most critically, we’ve clarified our Method section, added a Limitations section, and added several sections that deepened our analysis and conclusions about our corpus.  We have also added content that makes this contribution more germain to multimodal technology and cooperative intelligence.

We do have several questions. First, the link to the manuscript under “Please download the latest version of the manuscript for revision. Your original submission may have been changed.” Manuscript for Revisions takes us to a copy of the manuscript from July 12, 2022, well before our final submission. However, if we download the file “manuscript.pdf”, it seems correct (July 22, 2022 version).  We have updated the most recent version, but were puzzled about how this previous version got into the system.

Secondly, several reviewers made note of our Reference formatting. We used the BibTex (Overleaf) template, and we’re not sure how to change the predetermined formatting. Any assistance with this would be greatly appreciated.

Below is a list of each comment and how we addressed them.

Comment:

3: However, due to the study design,  this study made very little contribution 

to the body of knowledge in the field of self-driving. First, this study is highly descriptive. There is no theoretical contribution. Second, the context of this study is Tesla self-driving. The results of this study may not be applied to other  self-driving systems. For these reasons, this study cannot be seen as a research paper.

Response:

In response to this, we point out that we made the argument in detail that FSD is the first such system actually in widespread use, and so the only system that this kind of analysis could be done.  We have extended our discussion to further address how/whether this could be used by other self-driving systems, and in fact how the approach is general for many AI systems.

Round 2

Reviewer 3 Report

I want to thank the authors for their effort in revising the manuscript. The revised manucript is imporved. Again, this study is well-written and interesting to read. However, I still have the same concern that is the theoritical contribution of the study casued by the nature of study design.